# New Pre-reacted Glass Containing Dental Composites (giomers) with Improved Fluoride Release and Biocompatibility

**DOI:** 10.3390/ma12234021

**Published:** 2019-12-03

**Authors:** Loredana Colceriu Burtea, Cristina Prejmerean, Doina Prodan, Ioana Baldea, Mihaela Vlassa, Miuta Filip, Marioara Moldovan, Madalina-Anca Lazar (Moldovan), Aurora Antoniac, Vasile Prejmerean, Ioana Ambrosie

**Affiliations:** 1Faculty of Dentistry, Iuliu Hatieganu University of Medicine and Pharmacy, 8 Victor Babes Str., 400000 Cluj Napoca, Romaniaioanahodisan@yahoo.com (I.A.); 2Raluca Ripan Institute of Research in Chemistry, Babes Bolyai University, 30 Fintanele Str., 400294 Cluj Napoca, Romania; 3Department of Physiology, Iuliu Hatieganu University of Medicine and Pharmacy, 1 Clinicilor Str., 400006 Cluj Napoca, Romania; 4Department of Oral and Cranio-Maxillofacial Surgery, Iuliu Hatieganu University of Medicine and Pharmacy, 33 Motilor Str, 40001 Cluj-Napoca, Romania; 5Physical Metallurgy, Department of Metallic Materials Science, University Politehnica of Bucharest, 060042 Bucharest, Romania; 6Faculty of Mathematics and Computer Science, Babes Bolyai University, 7-9 Universitatii Str., 400084 Cluj Napoca, Romania

**Keywords:** dental giomers, residual monomer, fluoride release, cytotoxicity, SEM

## Abstract

The aim of the present work was to prepare a series of novel restorative giomers and investigate the morphology, the physico-chemical properties (residual monomer, fluoride release), and the cytotoxicity of the new materials. The experimental giomers were prepared as light-cured pastes by blending different resin matrices comprising aromatic/aliphatic/urethane (di) methacrylates, with hybrid fillers containing pre-reacted glasses (PRGs), a radiopaque glass, and nano fluorhydroxyapatite. Polyalkenoic acids based on acrylic acid/itaconic acid/N-acryloyl -L-leucine modified or not with methacrylic groups, together with a superficially active glass, were used to prepare the PRGs. The fluoride ion release of the experimental giomers was investigated within a period of 60 days of storage in bidistilled water while using a fluoride ion selective electrode. Beautifil II commercial product was used as a reference. Cell cytotoxicity tests were done in vitro, in accordance with ISO 10993-122012 proceedings. Human dermal fibroblasts and umbilical endothelial vein cultures were used. The values that were obtained for cumulative fluoride release for all experimental giomers were higher than for the Beautifil II product, being more than twice the ones that were obtained for the commercial product after 60 days of storage in bidistilled water. The experimental biomaterials showed similar and/or better results when compared to the commercial one; this effect was maintained in all tested conditions.

## 1. Introduction

The longevity and biocompatibility, along with the cariostatic effect, are essential in the clinical performance of the current aesthetic dental restorative materials.

Fluoride is well-known as an anticariogenic agent, with its most important action being the enhancement of remineralization of incipient caries lesions in enamel [1]. The release of fluoride ions from a dental material at the interface with an incipient dental caries leads to the formation of fluorapatite, which is more resistant compound than the hydroxyapatite. The demineralization process decreases and the remineralization process of the incipient caries begins through the formation of fluorapatite [2].

The most intensively studied fluoride-containing restorative materials that are available on the market include the products belonging to the classes of glass ionomer cements (GIC) and resin composites (RC) [3,4]. Conventional glass ionomer cements are materials made of calcium/strontium fluoroaluminosilicate glass powder combined with a polyalkenoic acid (water-soluble polymer) [5,6,7]. Besides fluoride release, conventional GICs present inherent (chemical) adhesion to tooth tissues, with a coefficient of thermal expansion similar to tooth structure and good biocompatibility, but they are sensitive to moisture and desiccation, have low wear resistance, low mechanical properties, and relatively poor aesthetics [8]. The resin-modified glass ionomer cements (RMGICs) present good fluoride release, but they are only a little more resistant than the conventional glass ionomers. Resin composites containing NaF or other soluble fluorides have been found to release fluoride ions and have an anticariogenic effect in vivo, but the elution of fluoride ions, in this case, would result in the loss of mechanical integrity of the restorative material [9,10].

Changes to the chemical composition of the above mentioned restorative materials were made in order to increase their performances. The incorporation of various amino-acids (glutamic acid, alanine, glycine, and proline) [11,12] in the polymer backbone of polyalkenoic acids (acrylic acid homo- or copolymers) was performed to increase the mechanical properties and also to promote better biocompatibility of the cured glass ionomers. The incorporation of hydroxyapatite/fluorapatite in the composition of GICs was done because of the same reasons [13]. Furthermore, the incorporation of hydroxyapatite or other nanomaterials in the composition of some light-cured dental unfilled and filled resins was found to confer enhanced mechanical properties to the biomaterials used in the dental field [14,15,16,17,18].

In the early 2000s, a new class of hybrid materials that combined the chemistry of resin composites with the one of the glass ionomer cements to obtain a controllable fluoride release was introduced on the market by Shofu (Kyoto, Japan), named giomers [19]. The giomers represent a new concept based on the Pre-Reacted Glass (PRG) technology [20], according to which fluoroaluminosilicate glass particles that have previously reacted with the polyacrylic acid are dispersed in the resin. The fluoride release from the giomer material continuously takes place [2,21] until the fluoride reservoir is depleted because of the extensive polysalt hydrogel matrix formed by the acid-base reaction. After that, the giomer can be recharged by fluorine containing toothpastes, mouthwashes, or other similar products, so that the fluoride re-release process continues [1,2]. Giomers were introduced in the class of smart materials due to their fluoride release and recharge abilities, as well as of their own action mechanism at the interface with the bone [21]. The in vivo clinical that were studies conducted for periods between three and 11 years reveal that the morphological, functional, and mechanical integrity of the giomer restorations is comparable to the one of the current resin composite restoratives [22,23,24].

The amount and nature of leachable components in giomers can influence their biocompatibility, with the smaller quantity of toxic substances eluted from these materials leading to improved results. The cytotoxic effect of giomers can be attributed to the release of unreacted monomers, such as triethylene glycol dimethacrylate and bisphenol A diglycidyl ether dimethacrylate, which are present in the material. Supplementary ions released from PRG fillers include aluminum, boron, sodium, silicon, strontium, [12,25], and zinc, which are the constituents of the fluoroaluminosilicate glass used in the PRG technology. The inhibition of enzyme activity, generation of ROS (reactive oxygen species), impairment of the antioxidant defense system, induction of inflammation, and apoptosis by low concentrations of fluoride in L929 mouse fibroblasts, [26,27] human dental pulp stem cells and humans gingival fibroblast have been reported [28].

Joining the constant concerns in the field, namely to obtain dental restorative materials with improved performances, our group previously developed certain experimental giomer materials and their corresponding adhesive systems, which were tested for monomer conversion [29], thermal properties [30], and marginal adaptation [31]. The marginal adaptation was evaluated by determining the microleakage of an experimental giomer in combination with two experimental two-step self-etch resin-modified glass-ionomer adhesive systems and then compared the results with the ones that were obtained for Beautifil II and FL-Bond II adhesive. The first original primer contained the polyalkenoic acid based on acrylic acid (AA) and itaconic acid (IA) modified with polymerizable groups as main components, while the second one contained a modified polyalkenoic acid, which contained the rest of N-acryloyl-L-leucine besides the AA and IA. The original giomer also contains a pre-reacted glass based on the second mentioned AA/IA/N-acryloyl-L-leucine copolymer. The study was conducted while using forty-two box-type class V standardized cavities that were prepared on the facial and oral surfaces of each tooth. After performing the clinical protocol, the restored teeth underwent 500 thermocycles in an LTC 100 thermocycling device (LAM Technologies Electronic Equipment, Firenze, Italy) that had two water baths at 5 °C and 55 °C. The dye penetration score and the percentage of the dye penetration length were determined. A lower microleakage value was recorded for the adhesive system that contained the AA/IA/N-acryloyl-L-leucine copolymer grafted with methacrylic groups as compared with the commercial adhesive.

Given the promising results that were obtained in our previous studies, in the present work we propose to prepare a series of novel experimental giomers with increased and prolonged fluoride release and improved biocompatibility. For this purpose, permeable resin matrices containing different combinations of (di) methacrylic monomers together with hybrid fillers comprising pre-reacted glasses based on polyalkenoic acids having amino acid moieties (L-leucine residue) with and without a low content of photopolymerizable groups in their structure along with fluorhydroxyapatite and a barium fluoroaluminoborosilicate glass were used as the main components in the original giomers.

## 2. Materials and Methods

### 2.1. Materials

2,2-bis [4-(2-hydroxy-3-methacryloxypropoxy) phenyl] propane (Bis-GMA), 1,6-bis (methacryloxy-2-ethoxy-carbonyl-amino) -2,4,4-trimethyl-hexane (UDMA), triethylene glycol dimethacrylate (TEGDMA), 2-hydroxyethyl methacrylate (HEMA), camphorquinone (CQ), dimethylaminoethyl-methacrylate (DMAEM), butylated hydroxy toluene (BHT), and 3-methacryloyloxypropyl-1-trimethoxy-silane (A-174 silane) were purchased from Sigma Aldrich Chemical Co. (Taufkirchen, Germany) and used without additional purification. The oxides and fluorides SiO_2_, Al_2_O_3_, ZnO, CaO, Na_2_O, B_2_O_3_, CaF_2_, and BaF_2_ were purchased from Merck. Beautifil II giomer—shade A30, batch number 051215, (PN1420 2015-04) was purchased from Shofu, Kyoto, Japan.

### 2.2. Preparation of Precursors

#### 2.2.1. Experimental Resins

The experimental resins were formulated while using monomer mixtures of Bis-GMA or UDMA as base monomers and TEGDMA or HEMA as the diluting monomers. The ratio between the base monomer and diluting monomer was 70/30. In the composition of the resins, besides the methacrylic oligomers and monomers, a photosensitizer, CQ in an amount of 0.5% (by weight), and an accelerator DMAEM, in an amount of 1% (by weight), were added. BHT was added in a quantity of 650 ppm related to the monomer mixture.

#### 2.2.2. Experimental Pre-reacted Glass Ionomer Fillers

The experimental pre-reacted glass ionomer fillers (PRG1 and PRG2) were prepared while using the conventionally method employed in the preparation of traditional glass ionomer cements. PRG1 was prepared by the hand-mixing of 50% aqueous solution of ternary copolymer P (AA-co-IA-co-Leu) obtained from acrylic acid (AA), itaconic acid (IA), and N-acryloyl-L-leucine (Leu) (molar ratio between AA/IA/Leu being 4:1:0.5) with the superficially active glass powder G having the oxidic composition SiO_2_ (49%), Al_2_O_3_ (22%), CaF_2_ (29%), in a weight ratio of 1/2.4. After seven days, the pre-reacted glass ionomer filler was dried in an oven (Memmert GmbH + Co.KG, Büchenbach, Germany) at 95 °C for 24 h. Finally, PRG1 was grounded in a ball mill (Retsch, Haan, Germany) and then sifted to fine powder. PRG2 was obtained in a similar manner; the only difference was that the modified copolymer P(AA-co-IA-co-LeuM) was used instead of P(AA-co-IA-co-Leu) copolymer. The methacrylic photopolymerizable groups were grafted on the copolymer backbone of P(AA-co-IA-co-LeuM) by the modification reaction of P(AA-co-IA-co-Leu) with methacryloyloxyethylcarbamoil-N’-2-hydroxyethylamine (degree of functionalization = 7%). The synthesis and characterization of the polyalkenoic acids P(AA-co-IA-co-Leu) and P(AA-co-IA-co-LeuM) were performed according to previously reported procedures [32]. The particle size distributions for the two glass powders were similar concerning d (0.1) (1.628 μm for PRG1 when compared to 1.584 μm for PRG2) and d (0.5) (10.760 μm in the case of PRG1 and 10.806 μm in the case of PRG2, respectively). PRG2 had a slightly higher d (0.9) than PRG1 (23.932 μm as compared to 21.992 μm) (Analysette 22 Nano Tec Laser-Particle-Sizer, FRITSCH, Idar-Oberstein, Germany) [29,33].

### 2.3. Obtaining of Experimental Light-curing Giomers

The experimental light-curing giomers were prepared as monopastes by mixing the resins with the hybrid fillers. For obtaining hybrid fillers, the pre-reacted glass ionomer fillers (PRG1 or PRG2), fluorohydroxyapatite (FHAP), and a silanized radiopaque glass having the composition SiO_2_ (25%), B_2_O_3_ (11%), Al_2_O_3_ (14%), BaF_2_ (50%) were mixed and then sifted together. The radiopaque glass powder was silanized with 3-methacryloyloxypropyl-1-trimethoxy-silane (A-174 silane). The volume median diameter of radiopaque glass powder was d_50_ = 5 microns (Analysette 22 Nano Tec Laser-Particle-Sizer) (FRITSCH, Idar-Oberstein, Germany) [26]. FHAP was used as nanopowder, in the form of rods with lengths between 15 and 160 nm and thicknesses of around 10 nm. The detailed method of obtaining and the characterization of the radiopaque glass and of FHAP are shown elsewhere [29,34].

The resin/hybrid filler ratio was 1/4. The composition of the experimental giomer pastes is presented in Table 1:

For obtaining hardened giomer samples, the giomer pastes were exposed for 20 seconds to a visible radiation having the wavelength in the range of 470 nm and the intensity of 950 mW cm^−2^ generated by LED.E dental lamp (Guilin Woodpecker Medical Instruments Co., Guangxi, China).

### 2.4. Methods of Characterization

#### 2.4.1. Residual Monomer

The method of evaluating the quantity of unreacted monomer that is proposed in the present work is based on the accelerated extraction of the eluted components from the samples in chloroform containing 0.01% BHT.

The disc samples (10 mm in diameter and 1 mm in height) were light cured, weighed, and, immediately after this, they were subjected to the extraction of the residual monomers in 50 mL chloroform at 60 °C for 10 h. Our group developed this method of extraction of the unreacted monomer from dental composites and it was used in certain previous studies [35]. The each extraction solution was evaporated to dryness and the residue was re-suspended in 2 mL of acetonitrile. The samples were diluted at an appropriate concentration, filtered with 0.45 mm polytetrafluoroethylene (PTFE) filters, and then analyzed by High Performance Liquid Chromatography (HPLC).

The analyses were carried out on a Jasco Chromatograph (Jasco International Co., LTD., Tokyo, Japan) that was equipped with an intelligent HPLC pump (PU-980, Jasco International Co., LTD., Tokyo, Japan), a ternary gradient unit (LG-980-02, Jasco International Co., LTD., Tokyo, Japan), an intelligent column thermostat (CO-2060 Plus, Jasco International Co., LTD., Tokyo, Japan), an intelligent UV/VIS detector (UV-975, Jasco International Co., LTD., Tokyo, Japan), and an injection valve that was equipped with a 20 μL sample loop (Rheodyne, Thermo Fischer Scientific, Waltham, MA, USA). The system was controlled and the experimental data analyzed with the ChromPass software (version v1.7, Jasco International Co., LTD., Tokyo, Japan). Separation was carried out on a Lichrosorb RP-C18 column (25 × 0.46 cm) at 21 °C column temperature. The mobile phase was a mixture of acetonitrile (A, HPLC grade) and water (B, Milipore ultrapure water), and a gradient was applied according to the following method: 0–15 min. linear gradient 50–80% A; 15–25 min. linear gradient 80–50% A. The flow rate was 0.9 mL/min. and the injector volume was always 20 μL. UV detection was performed at 204 nm for monitoring the elution of all analytes (BisGMA, TEGDMA, HEMA, and UDMA), because they exhibit significant absorption at this wavelength. Stock solutions of reference standards of Bis-GMA, TEGDMA, and UEDMA (1 mg/mL) were prepared in acetonitrile were stored at 4 °C. The linearity of the response to analytes was established with four concentration levels and the regression factors R^2^ were higher than 0.998. Five samples were investigated for each giomer composition. The residual monomer amount has been determined from the HPLC chromatograms of the extracts and it was calculated as percent related to the initial amount of the monomer in the sample and related to the weight of the sample, respectively.

#### 2.4.2. Fluoride Release

Disc samples of giomers measuring 15 mm in diameter and 1 mm in thickness were made in teflon molds by exposure to visible radiation generated by LED.E dental lamp for 20 seconds of five points on the disc surface. The samples were suspended in 45 ml bidistilled water (Simplicity UV, Water Purification System Millipore, Santa Ana, CA, USA)/5 mL TISAB III buffer (Total ionic strength adjustment buffer concentrate solution, HI 4010-06, Hanna Instruments, Woonsocket, RI, USA) at 37 °C. Fluoride analysis was carried out while using a fluoride ion selective electrode (Combination Fluoride Electrode HI 4110 filled with HI 7075 equitransferent electrolyte for reference electrodes, Hanna instruments, Woonsocket, RI, USA) connected to a pHmeter (Hanna HI 221 pH Meter with calibration check). After each measurement, the samples were placed in the same corresponding vials and kept in a thermostatic bath at 37 °C. The meter/electrode was calibrated while using a series of standards of known concentration from 10^−5^–10^−2^ mol/L F^−^. The standard solutions are used to plot the calibration graph. The measurements for the standards and the sample solution were made in 50 mL bidistilled water/TISAB III buffer (45:5) at 37 °C (±2). All experiments were performed in polyethylene labware. Seven samples were measured for each giomer composition. Fluoride release was recorded in ppm.

#### 2.4.3. Flexural Strength

The flexural strength was determined while using specimens having rectangular form (length 25.0 mm, height 2.0 mm, and width 2.0 mm), according to ISO 4049/2000 [36]. The specimens were light cured in their respective molds while using the LED.E dental lamp as light source, in five points along the length of the specimen, on both sides, for 20 seconds in each point. The assembly was placed in the water bath for 37 °C for 15 min. Subsequently, the specimens were removed from the molds and after the measurement of height *h* and width *b* and they were subjected to three-point loading with l = 20 mm between the supports. The measurements of the flexural strength were made using a Lloyd LR5K Plus mechanical testing apparatus. The crosshead speed of the testing machine was 0.75 mm/min. For the three-point flexural strength test, five specimens were fabricated from each giomer material.

The flexural strength (FS) was calculated as
(1)$$\mathrm{FS}=3\mathrm{FI}/2{\mathrm{bh}}^{2}$$

FS is the flexural strength (in MPa), F is the maximum load applied to the specimen (N), l is the span between the supports (20 mm), and b and h are the width and height, respectively, of the specimen in mm.

#### 2.4.4. Scanning Electron Microscopy (SEM)

Disk-shaped samples (8 mm diameter, 4 mm high) were prepared for the determination of surface morphology of the cured giomers. The giomer samples were embedded in poly (methyl methacrylate) (PMMA) and they were then transversely sectioned in slices of 1 mm thickness while using a diamond saw (Isomet 1000, Buehler, IL, USA). All of the samples were SiC papers polished using 800, 1000 SiC abrasive paper. The investigations were performed using Quanta 3D FEG (FEI Company, Hillsboro, OR, USA) equipment.

The fractured surface of the samples was evaluated using Inspect S (FEI Company, Hillsboro, OR, USA) apparatus in the low vacuum mode for the investigation of the fracture appearance of the giomer samples after the flexural test.

#### 2.4.5. Cytotoxicity Assay

Cell cultures: Human dermal fibroblasts HDFa (Invitrogen, Willow Creek, CA, USA) and human umbilical endothelial vein cultures (HUVEC) (Promocell, Hamburg, Germany) were used. Fibroblasts were cultured in Dulbecco’s modified Eagle’s medium (DMEM), while HUVEC’s were cultivated in RPMI medium, both supplemented with 5% fetal calf serum, 50 µg/mL gentamicin, and 5 ng/mL amphotericin (Biochrom Ag, Berlin, Germany). The cytotoxicity assay evaluated the overall effect of the biomaterials on normal, human cell lines in vitro. Human dermal fibroblasts, as well as mouse 3T3 fibroblasts [37] or L929 fibroblasts [38], were previously employed for dental materials testing [39,40,41], since the cytotoxic effect is similar to that exerted on oral fibroblasts, while the HUVECs [42] were used for comparison, because they are fetal cells that can mimic the stem cells behavior when exposed to eluted substances from the tested biomaterials.

The biomaterial extracts were obtained in compliance with the ISO 10993-12:2012 proceedings. A thin sample (0.5–1 mm) of each material ≈ 3 cm^2^/surface area was incubated completely submerged in 1ml of culture medium for 24 and 72 h at 37 °C. Six samples were incubated for each giomer composition. The extracts were immediately used for cell viability assays.

Cytotoxicity assay: The cells seeded at a density of 10^4^/well in ELISA 96 wells micro titration flat bottom plaques (TPP, Trasadingen, Switzerland) were allowed to settle for 24 h. The cells were then directly exposed for 24 h to extracts of each biomaterial sample (prepared as above) and diluted in a range of 1–0.001 in cell culture medium. Afterwards, cells were washed and viability was measured by the colorimetric measurement of formazan, a coloured compound generated by viable cells while using CellTiter 96^®^ AQueous Non-Radioactive Cell Proliferation Assay (Promega Corporation, Madison, WI, USA), and readings were done at 490 nm using an ELISA plate reader (Tecan Austria GmbH., Grödig, Austria). Each experiment was done in triplicate. The results are presented as the average of the percents of untreated control absorbance (OD490). Untreated cultures exposed to medium were used as controls.

#### 2.4.6. Statistical Method

The statistical significance of the difference in fluoride release and cell viability between the commercial material Beautifil II, and the experimental materials were evaluated by one-tail, paired Student TTEST, using GraphPad; results were considered to b significant for *p* ≤ 0.05 (level of significance α = 0.05). Statistical package Prism (version 4.00 for Windows, GraphPad Software, San Diego, CA, USA) www.graphpad.com was used for data analyses.

## 3. Results

### 3.1. Residual Monomer

The HPLC chromatograms of the extracts of the A1, A2, and A3 and Beautifil II samples were recorded, together with the HPLC chromatogram of the standard mixture of monomers (Figure 1), and the amount of the residual monomer was calculated in each case.

Table 2 and Table 3 show the percentages of the Bis-GMA, UDMA, HEMA, and TEGDMA unreacted monomers that were extracted in chloroform at 60 °C for 10 h (the maximum concentration being reached after eight hours under these conditions) related to the initial amount of the corresponding monomer in the sample and to the weight of the sample, respectively.

As can be seen from the data in Table 2, the highest percentage of unreacted Bis-GMA monomer was recorded in the case of A2 (2.13 %), followed by B1 (1.65%), A1 (1.62%), and Beautifil II samples (1.08%). In the case of A3, the percentage of unreacted UDMA (0.83%) was lower than that of unreacted Bis-GMA in the case of all other samples (A1, A2, A3, and Beautifil II).

Regarding the dilution monomers TEGDMA and HEMA, the highest amount of unreacted TEGDMA was registered for B1 (0.70%), but this percent is smaller than the amount of unreacted HEMA in the case of A2 (3.65%).

The same mentioned correlations can be deduced from the data that are presented in Table 3, in addition one can observe the total residual monomer (cumulative residual monomer), ranging from 0.764% in the case of Beautifill II and 2.59% in the case of A2 sample.

### 3.2. Cumulative Fluoride Release

Figure 2 shows the cumulative fluoride release data (mean values and standard deviations) for A1, B1, A2, A3 experimental giomers, and Beautifil II commercial product within a period of 60 days of storage in bidistilled water.

The values that were obtained for cumulative fluoride release for all experimental giomers were higher than for Beautifil II product, being more than twice the ones that were obtained for the commercial product after 60 days of storage.

Statistical analysis indicated significant differences in fluoride release between Beautifil II and the experimental giomers during the experiments. The highest differences were obtained in the case of comparing Beautifil II with A3 giomer (*p* = 1.429 × 10^−6^ for one day; *p* = 2.29 × 10^−7^ for one week, *p* = 7.23 × 10^−7^ for one month, *p* = 3.128 × 10^−6^ for two months of storage), while the smallest differences were recorded between Beautifil II and A1 giomer (*p* = 4.131 × 10^−5^; *p* = 1.2888 × 10^−3^, *p* = 4.322 × 10^−4^, *p* = 4.414 × 10^−5^ for the same periods of time). Between Beautifil II and A2 and B1, respectively, the statistical differences were also significant (*p* ≤ 3.43 × 10^−5^ and *p* ≤ 2.66 × 10^−3^, respectively).

The differences in cumulative fluoride release were not statistically significant for one day and two months of storage (*p* = 0.5 for one day and *p* = 0.256 for two months, respectively) when comparing the A1 giomer with the B1 giomer, however the differences in cumulative amount of leached fluoride were significant for one week and for one month (*p* = 0.035 and *p* = 0.0121, respectively).

If the average cumulative fluoride release of the A1, B1, A2, and A3 experimental giomers and the cumulative fluoride release of Beautifil II commercial product, respectively, were plotted against time, a linear correlation between the cumulative release and time was found. The linear regression analysis of the data provided the correlation coefficients of 0.998 in the case of experimental giomers and 0.9847 in the case of Beautifil II (Figure 3).

### 3.3. Flexural Strengths

Figure 4 presents the flexural strengths of the experimental A1, B1, A2, and A3 giomers and of the commercial Beautifil II.

The values of the flexural strengths of the experimental giomers ranged between 89.2 MPa for A2 giomer and 108.8 MPa for the A3 giomer. Beautifil II presented the highest value (115.7 MPa). All of the giomers presented values of flexural strengths above 80 MPa, the limit the international standard ISO 4049/2000 imposed.

### 3.4. SEM Investigations of Giomers

Figure 5 shows the surface morphology and fracture image after the flexural test (in the upper right corner) of A1, B1, A2, A3, and Beautifil II giomers.

A1, A2, A3, and B1 giomers present a similar surface organization. Figure 5a–d present a large amount of particles with sharp or rounded edges (shapes) measuring less than 10 microns as well as a few particles having a diameter of about 20 microns. The surface morphology of Beautifil II shows particles of various sizes, mostly below 15 microns also having irregular shapes (Figure 5e). One can observe a greater amount of particles embedded in the polymer matrix than in the case of experimental giomers. The black pores on the samples surface in the SEM images of all the giomers appeared due to the preparation of the samples (sectioning and sanding) during which particles of different sizes may detach from the polymer matrix.

### 3.5. Cell Cytotoxicity

Figure 6 presents the viability of fibroblasts and of HUVECs exposed to giomer extracts for 24 h and 72 h, respectively.

When the undiluted extracts were used, the viability of fibroblasts ranged between 32.27% (Beautifil II) and 52.02% (A3) after 24 h of extraction, and between 37.88% (A1) and 47.70 (A3) after 72 h of extraction. When HUVEC were used, the viability ranged from 60.40% (A2) to 99.34% (B1) after 24 h and from 45.22% (A2) to 99.57% (Beautifil II) after 72 h of extraction.

All of the tested biomaterials decreased cell viability in a dose related manner, in both cell lines (Figure 6). The overall viability of the fibroblasts (Figure 6a,b) decreased more when compared to that of the HUVECs (Figure 6c,d).

The time used for the sample extraction (24 or 72 h) had a different impact on cell viability depending on the materials used. As such, viability of the cells that were exposed to the commercial material Beautifil II decreased more at 24 h (*p* = 3.73 × 10^−10^ for fibroblasts, *p* = 3.15 × 10^−10^ for HUVEC) than at 72 h (*p* = 3.57 × 10^−9^ for fibroblasts, *p*= 3.31 × 10^−1^ for HUVEC) when compared to untreated controls. The experimental materials exerted a different effect. They induced a more important decrease at 72 h (*p* ≤ 2.62 × 10^−7^ for fibroblasts and *p* ≤ 7.79 × 10^−5^ for HUVEC) than at 24 h (*p* ≤ 4.73 × 10^−6^ for fibroblasts and *p* ≤ 3.74 × 10^−5^ for HUVEC, however, *p* = 1.1 × 10^−1^, not significant for A1) as compared to controls.

When the experimental giomers were compared with the commercial material Beautifil II, significant differences in viability of fibroblasts were recorded at 24 h (between *p* = 5.72 × 10^−8^ for A1 and *p* = 2.14 × 10^−5^ for A3) and at 72 h (between *p* = 3.67 × 10^−4^ for A3 and *p* = 2.59 × 10^−2^ for B1), however, *p* = 4.58 × 10^−1^ not significant for A2. The differences between the commercial product and the experimental materials were higher when HUVECs were used, these ranged between *p* = 8.22 × 10^−10^ (A1) and *p* = 1.68 × 10^−6^ (A3) at 24 h and *p* = 8.40 × 10^−10^ (A2) and *p* = 5.31 × 10^−6^ (B1) at 72 h.

Human fibroblasts and HUVECs both showed a constant dose-dependent viability increase when exposed to diluted conditioned medium, reaching, in some cases, to overcome 100% cell viability (A1, B1, and Beautifil II).

## 4. Discussion

### 4.1. Residual Monomer

The giomers harden through the radical polymerization of the monomers from the resin matrix. A crosslinked three-dimensional polymer network is formed, as in the case of the light-cured resin composites. The polymerization rate is different at different points in the network during the formation of the crosslinked network, thus being higher in the so-called "microgel" regions due to the appearance of the local gel effect. The gel effect determines the decreasing of the diffusion rate of macroradicals and unreacted monomers in these regions, leading to their occlusion in the crosslinked network. In this way, a significant percentage of the double methacrylic bonds remain unreacted as pendant methacrylate groups attached to the polymer network or as free residual monomer, preventing the complete conversion of the monomers [9,44,45,46].

It is desirable that the residual monomer remain as small as possible as a result of the polymerization of the methacrylic monomers, because the unreacted monomer can be extracted from the giomer material in saliva, this fact having a potential impact on both the structural stability of the dental material, as well as on its biocompatibility. The residual monomers can be extracted into the salivary fluid and brought into contact with the mucosal tissue. Furthermore, the components can be extracted into dentine, from where they can diffuse into the dental pulp and cause irritation in the dentin.

The discovery that a proportion of the initial monomer remains unreacted during the hardening of the diacrylic dental resins, as it is found as a residual monomer trapped in the polymeric matrix, has led a number of researchers to study the phenomenon of elution of these unbound molecules in different cured composite materials [47,48,49,50,51,52,53]. The studies have shown varied values of elution in water in the range of 0.05 to 2% of the weight of the specimen and higher values of elution in the organic solvents (2–11%). Inoue and Hayashi [47] have suggested that only 10% of the unreacted monomer is elutable in water, while Ferracane found that about 10% of the total unreacted methacrylate double bonds was elutable as residual monomer. In water, most of the unreacted monomer elutes within 1–3 days, and only Bis-GMA elutes for weeks [50].

Even if there are relatively numerous studies that present the evaluation of the extractable products and the residual monomer from the conventional composite resins in the literature, in the particular case of giomers, according to our knowledge, there are no studies regarding both the evaluation of leachable components and of the residual monomer.

In our experiments chloroform (which has a solubility parameter range around 19.03 J^1/2^ cm^−3/2^ was used as the extraction solvent in order to evaluate as precisely as possible the amount of unreacted monomer from the investigated giomers. In this way it was possible to more accurately quantify the unreacted monomers (Bis-GMA, TEGDMA, UDMA, and HEMA) resulting from the photo-polymerization reaction by avoiding the differences between the solubility of the monomers in aqueous media [54]. Furthermore, the extraction of the unreacted monomers took place at 60 °C to accelerate the elution of the unreacted species. In these conditions, both the experimental samples A1, B1, A2, and A3 and Beautifil II product released residual monomer in an amount less than 3% of the initial weight of the sample, with this value being smaller than the value obtained for the resin composites that were extracted in chloroform (7.5%) [49].

When comparing the A1 and A2 specimens that contain Bis-GMA as a base monomer, a greater amount of residual Bis-GMA was recorded in the case of the A2 sample (2.13% of the initial Bis-GMA amount in the sample) than in the case of A1 sample (1.62%). This result suggests that a network with a higher degree of conversion of the methacrylate double bonds is formed by the polymerization of Bis-GMA with TEGDMA than by polymerization of Bis-GMA with HEMA. The fact that the obtained value for the residual HEMA was much higher (3.67%) than the value that was obtained for the residual TEGDMA (0.61%) confirmed the finding.

A1 and B1 giomers, which contain the same resin (Bis-GMA and TEGDMA) and different PRGs, present similar values for the residual monomer. This result confirms the findings that were obtained in a previous study, according to which the degree of conversion in giomers only depends on the composition of the resin, being unaffected by the filler [29].

In the case of A3 sample, smaller amounts of unreacted UDMA and HEMA, respectively, were recorded (0.83% and 1.17%, respectively), suggesting a higher degree of conversion, which can be explained by the lower viscosity of the resulting material that leads to a longer gel time and consequently to a greater number of reacted double bonds.

### 4.2. Fluoride Release

The major mechanism of fluoride ion release in giomers is an ion exchange process, with a fluoride ion being exchanged for a hydroxyl ion [55]. It is speculated that the PRG fillers promote fluoride release through a ligand exchange within the pre-reacted hydrogel [10,19].

The relatively scarce data presented in literature regarding the properties of the commercial restorative giomers (Beautifil and Reactmer—the first generation products and Beautifil II—the second generation giomer) show different values for the amount of released fluoride ions. The data showed differences between 5.95 μg/cm^2^ after 21 days of storage to 9.56 μg/cm^2^ after 10 weeks for Beautifil, 23.56 μg/cm^2^ after 10 weeks for Reactmer, and 5.87 ppm after 28 days for Beautifil II [1,56,57,58]. This lack of correlation could be explained by the different sizes of samples (between 4 mm and 10 mm in diameter), the different amounts of storage media (between 1 mL and 15 mL deionized water), as well as by the regime of the samples’ transfer to a new storage medium (between one and 31 days) and the time of storage used in the studies. The cumulative fluoride release was higher when the storage medium volume/sample diameter ratio was higher [56,59]. The values for the cumulative fluoride release of the commercial giomers that were shown in the literature were lower than the ones that were obtained in our study for Beautifil II. Moreover, our results suggested that there is a linear correlation between the fluoride release and the time of storage, following the conditions of our experiment (60 days of storage in the same medium, a large volume of storage–45 mL bidistilled water–and a relative large diameter of the sample–15 mm), without an initial burst in the first days (Figure 3).

The concentrations of fluoride ions released by the experimental giomers A1, B1, A2, and A3 at day 60 are comparable to the amount of fluoride ions leached by glass ionomer cements. In conventional glass ionomers, after an initial burst of 15–155 ppm in the first day, the fluoride release decreased to about 0.9–4 ppm at day 60 [60]. In the case of the resin modified glass ionomers, the initial release of 8–15 ppm from the first day dropped to 1–2 ppm on the seventh day [61,62], value that lasts longer (1–2.7 years, depending of the sample size) [2].

The highest value for the cumulative fluoride release was obtained in the case of the A3 giomer when comparing the A1, A2, and A3 experimental giomers that contain the same hybrid filler based on PRG1 and different resin matrices (0.47 ppm after one day; 9.26 ppm F^−^—after one week, 44.48 ppm F^−^—after one month, and 84.43 ppm F^−^—after two months). The A3 giomer contained the most hydrophilic and flexible polymer matrix based on UDMA aliphatic urethane dimethacrylate and HEMA. The polymer matrix of the A2 and the A1 giomers based on more rigid and hydrophobic aromatic dimethacrylate Bis-GMA and HEMA, and on Bis-GMA and TEGDMA, respectively, led to obtaining of lower values for the cumulative fluoride release—after 60 days of storage, the values were 77.92 ppm F^−^ and 67.44 ppm F^−^, respectively.

The values recorded for the A1 and B1 giomers, which contained the same resin (Bis-GMA/TEGDMA), but different PRG (PRG1 and PRG2, respectively), are very close throughout the measurements, with B1 releasing a constantly slightly larger amount of fluoride ions than A1 all the time. It is possible that during the polymerization of the resin, which involved the methacrylic groups from P(AA-co-IA-co-LeuM), more hydrophilic and permeable domains surrounding the PRG2 particles were formed and so an increased fluoride release could have occurred.

When comparing the experimental giomers with Beautifill II, the commercial product presented the lowest values: 0.07 ppm after one day; 2.59 ppm F^−^—after one week, 13.31 ppm F^−^—after four weeks, and 33.56 ppm F^−^—after 60 days. This behavior could be explained by the fact that by using the amino acid modified polyalkenoic acids in the composition of the PRGs in the experimental giomers instead of the polyacrylic acid from the PRG in Beautifil II, the steric hindrance between the carboxyl groups that are directly attached to the backbone and are closely oriented to each other could be reduced, leading to the possibility of more carboxyl groups undergoing reactions with the cations released from the superficially active glass. As a consequence, more fluoride ions could be released from the experimental giomers.

Analyzing the average amount per day of fluoride ions released in bidistilled water in the first week, all of the giomers, except A3, presented lower concentrations than 1 ppm/day. These concentrations increased during the second week when the experimental giomers showed a mean concentration that ranged from 0.96 ppm/day (A1) to 2.46 ppm/day (A3). Beautifill II released an average of 0.42 ppm fluoride ions/day in the second week when compared to 0.36 ppm/day in the first week. Starting with the third week the mean concentrations slightly increased in the case of Beautifill II, A1 and B1 giomers, and slightly decreased in the case of A2 and A3 giomers until the eighth week. The amount of leached fluoride ions per day became similar for the experimental giomers on day 60, ranging from 1.5 ppm to 1.64 ppm, while Beautifil II presented half of the experimental giomers concentration (0.83 ppm).

### 4.3. Flexural Strengths

The filler component of the material influenced the mechanical properties of dental composites. The size, shape, loading, distribution, and silanation of the filler particles are related to the values of the mechanical strengths of a certain composite material. However the role of the chemical composition of the resin should not be neglected, because it determines the conversion of the monomers and the level of cross linking of the tridimensional polymer network [63,64].

The highest flexural strength value was recorded in the case of the A3 giomer based on the UDMA/HEMA resin (108.8 MPa) when comparing the experimental monomers A1, A2, and A3 that contain the same hybrid filler and different resin matrices. The A3 giomer contains UDMA as base monomer, which has the molecular weight close to that of Bis-GMA, but it is much more flexible than Bis-GMA (the viscosity of Bis-GMA is about 1,000,000 mPa.s (23 °C) as compared to the viscosity of UDMA of 11,000 mPa.s) [9]. The values that were recorded for the flexural strengths confirm the results that were obtained in the case of evaluating the residual monomer and also suggest obtaining a higher conversion and cross linking density in the A3 giomer based on UDMA than in the A1 and A2 giomers based on Bis-GMA. These findings are in agreement with other studies that showed that the UDMA-based resin composites present improved mechanical properties when compared to the Bis-GMA–based composites [6,9].

A slightly higher flexural strength value was recorded in the case of B1 giomer (99.8 MPa) than in the case of A1 giomer (97.9 MPa) if we compare the A1 and B1 giomers that contain the same resin based on Bis-GMA/TEGDMA and different PRGs. This is probably due to the modified surface of the PRG2 with methacrylic groups that can polymerize with the methacrylic groups of the monomers, leading to a strong bond between the PRG2 and the B1 polymer matrix.

Beautifil II presented the highest flexural strength value when compared to all of the experimental giomers (115.7 MPa). The higher amount of filler particles in the commercial giomer than in the experimental materials can explain this value. Al_2_O_3_ can also have a beneficial influence upon the flexural strength value of Beautifil II [64].

### 4.4. SEM Investigations of Giomers

Nowadays, SEM is a useful tool in the investigation of the microstructure of biomaterials used in dentistry [65]. The microstructural characteristics of dental composites, in particular case of dental giomers, exert a decisive influence on the properties of the materials [63].

We investigated the surface and fracture morphology of the experimental giomers A1, B1, A2, and A3 to compare it with the one of the commercial product Beautifil II to detect any differences in size, shape, distribution, or amount of the filler particles that could influence the properties of the investigated giomers, in particular the fluoride release and flexural strength.

The particles measuring less than 10 microns can be attributed to the radiopaque filler particles or small sizes PRG1 (PRG2) particles and the second ones (measuring about 20 microns) to the large sizes PRG1 (PRG2) filler particles while analyzing the SEM photomicrographs of the surface morphology of experimental giomers presented in Figure 5a–d and based on the particle size analysis. One can observe the distribution of the PRG1 (PRG2) filler and the other particles belonging to the radiopaque filler well embedded in the polymer matrix.

Beautifil II commercial product presented smaller particles than the experimental giomers having less than 15 microns. Beautifil II presents a higher filling load than the experimental materials, which is consistent with the composition of Beautifil II shown in the literature (S-PRG fillers 68.6% w/v, 83.3% w/w, Bis-GMA, TEGDMA) [1] or in the “Material Safety Data Sheet” (Bis-GMA 7.5%, TEGDMA < 5%; Aluminofluoro-borosilicate glass 70, Al_2_O_3_) [43]. A compact packing of the filler particles in the polymeric matrix was observed for all of the materials.

The SEM photomicrographs of the fracture surface of the investigated giomers reveal the same homogeneity of the material inside of the samples as in the case of surface morphology, which highlights the very good interfacial adhesion between the filler particles and polymer matrix. There are no additional cracks or dislocated particles present at the fracture surface. Based on the cracking cross-section from Figure 5c, it can be assumed that the initial crack occurred in the polymer matrix as a result of the existence of an initial microvoid or hole in the material, and it then grew rapidly due to the applied force, finally leading to the material breaking.

The fluoride release of Beautifil II was smaller than the one that was obtained in the case of experimental giomers, even though Beautifill II contains a greater amount of filler particles than the experimental giomers. Undoubtedly, this behavior can be explained first of all by the different composition of the investigated materials. However, the smaller reinforcement of the experimental giomers, which allows for a greater segmental mobility of the polymer chain, can contribute to the greater release of fluoride ions in the case of the experimental giomers [10].

### 4.5. Cytotoxicity

The cytotoxicity of resin-based dental materials was evaluated in the literature while using different primary and immortalized cell lines, with the obtained results being influenced by the used cell type [66]. So far, there is no literature consensus regarding the most suitable cell type for the evaluation of cytotoxicity in vitro of some types of biomaterials, in particular of dental products [67,68,69,70].

We tested the cytotoxicity of the four experimental giomers and of the commercial product Beautifil II on two human normal cell lines, which were previously used for the study of dental materials: fibroblasts [40,41] and endothelial cells [42], according to ISO 10993-5 (1992, Biological Evaluation of Medical Devices-Part-5: Tests for cytotoxicity-in vitro methods).

When the undiluted extracts were used, all of the giomers exerted a decrease in cell viability in both tested cell lines (Figure 6). The results are in agreement with the literature data in which there is a general consensus that resin-based dental materials are cytotoxic mainly at early intervals [71], but they are different from the results that were obtained by Reza Pourabbas et al. who didn’t find any cytotoxicity for Beautifil R giomer [72].

A3, B1, and A1 had a lower effect on the fibroblasts, in the case of HUVEC’s, B1 and A1 induced a very slight decreased of the viability when comparing the effects of the experimental giomers upon the two cell lines. HUVEC’s viability was practically undisturbed after 24 h of exposure. This could be explained by the fact that fibroblasts are adult cells with reduced proliferative capacities, which makes them more suitable for biomaterials testing.

Besides the type of the cell line used, the cytotoxic effect was influenced by the composition of the material and the preparation time of the extract. According to the literature data, the commercial giomer Beautifil II has the same Bis-GMA and TEGDMA monomers in its composition as the A1 and B1 experimental giomers [43]. Although the content of Bis-GMA in Beautifil II is about half of the content in A1 and B1 and the content of TEGDMA is almost the same, the citotoxicity induced by Beautifil II in both tested cell lines is noticeably higher than the one induced by A1 and B1 at 24 h. Conversely, after 72 h, the viability of cells that were exposed to Beautifil II increased while the viability of cells that were exposed to A1 and B1 decreased, so that, at this time, the citotoxicity of Beautifil II became smaller than the one of A1 and B1, with the effect being more pronounced in the case of HUVEC’s. The behavior of Beautifil II could probably be due to the fast elution of other compounds than the residual monomers from its composition with cytotoxic potential (ions, filler particles, or additives) within the first 24 h, followed by their transformation in chemical species with lower cytotoxic potential in the next 48 h.

It appears from our results that a release of fluoride ions of up to 2.5 ppm per day does not lead to cytotoxic effects on the two cell lines studied regarding the presumed cytotoxicity induced by the higher concentration of fluoride ions released in the case of experimental giomers.

When comparing the A1 and B1 experimental giomers, the induced cytotoxicity upon both of the tested cells was almost similar and the differences in the cell viability were smaller than 5%. This shows that the modified polyalkenoic acid in PRG2 did not induce a significant toxicity as compared to the unmodified polyalkenoic acid in PRG1.

There was no significant difference between the A1 and A2 giomers upon fibroblasts viability, which suggests that TEGDMA and HEMA induced a similar cytotoxic effect upon the fibroblast cells. This behavior is in accordance to other findings [66]. A different behavior was observed in the case of exposing the A1 and A2 giomers to HUVEC, when the viability was drastically decreased in the case of A2. This could be due to the increased cytotoxic effect of HEMA upon the HUVEC compared to the one induced by TEGDMA.

With respect to the A2 and A3 experimental giomers, which differ only by the nature of main monomer (Bis-GMA vs. UDMA), the obtained results show that the cytotoxic effect of A2 is higher than of A3. This behavior could be explained by the higher conversion of the polymerization and, consequently, by the lower content of residual monomer in A3 as compared to the A2 giomer.

## 5. Conclusions

The results that were obtained in the present work reveal that the composition of the resin matrix and of the pre-reacted glass substantially influences the physico-chemical properties (residual monomer, fluoride release) and biocompatibility of the giomers.

Our findings suggest that the use of an enhanced hydrophilic resin matrix and of polyalkenoic acid with L-leucine residue as precursor for the pre-reacted glass leads to an increased and prolonged fluoride release. In addition, the presence of an amino acid moiety in the PRG leads to obtaining giomer materials having improved biocompatibility.

In consequence, the giomer materials comprising polyalkenoic acids with L-leucine residue can be promising candidates for dental applications.

## Figures and Tables

**Figure 1 materials-12-04021-f001:**
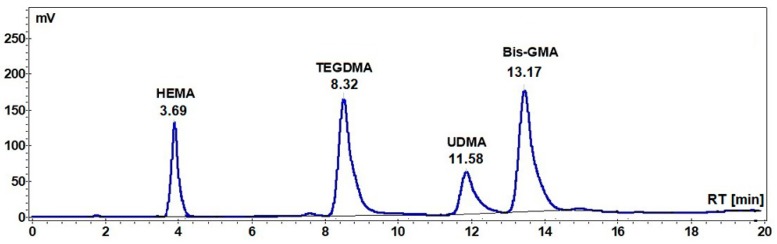
High Performance Liquid Chromatography (HPLC) chromatograms of the standard mixture of monomers (30 μg/mL).

**Figure 2 materials-12-04021-f002:**
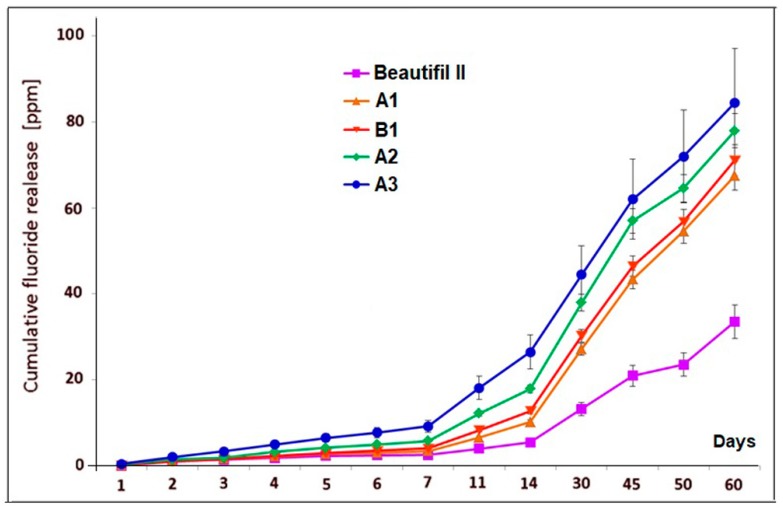
Cumulative fluoride release of commercial Beautifil II and experimental A1, B1, A2, and A3 giomers.

**Figure 3 materials-12-04021-f003:**
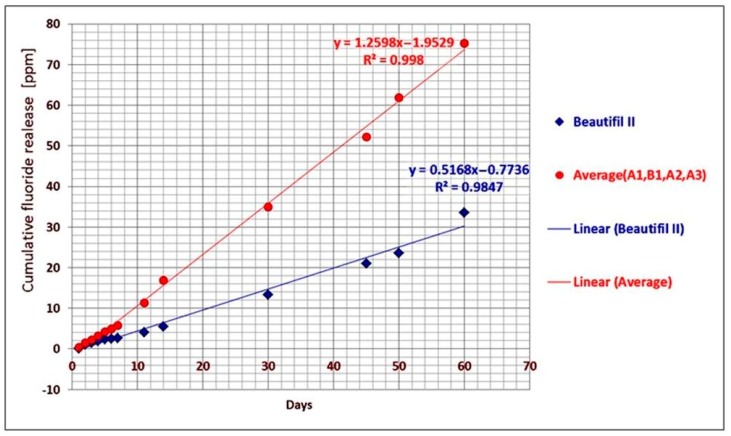
Average cumulative fluoride release of **A1**, **B1**, **A2**, and **A3** experimental giomers and of Beautifil II commercial product plotted against time.

**Figure 4 materials-12-04021-f004:**
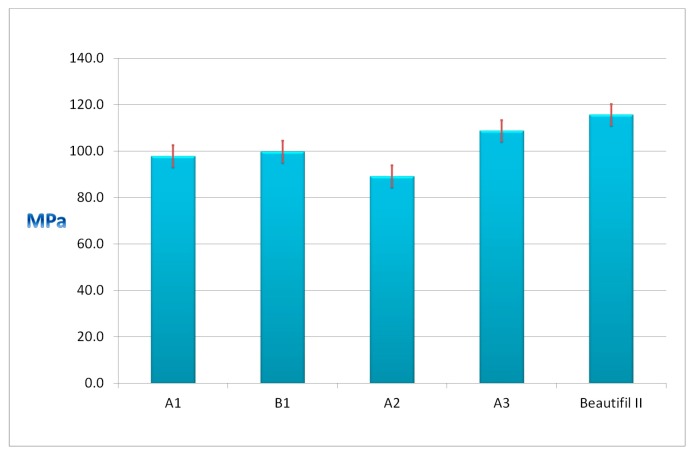
The flexural strengths of the investigated giomers.

**Figure 5 materials-12-04021-f005:**
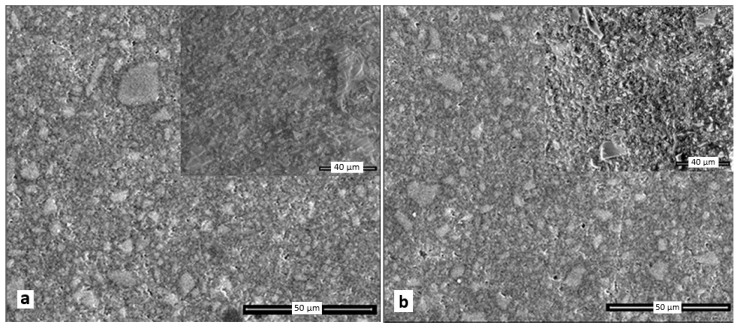
Scanning Electron Microscopy (SEM) photomicrographs of (**a**) A1; (**b**) B1; (**c**) A2; (**d**) A3; and, (**e**) Beautifil II giomers. Surface morphology (×1000) and fracture image after the flexural test (×2000) in the upper right corner.

**Figure 6 materials-12-04021-f006:**
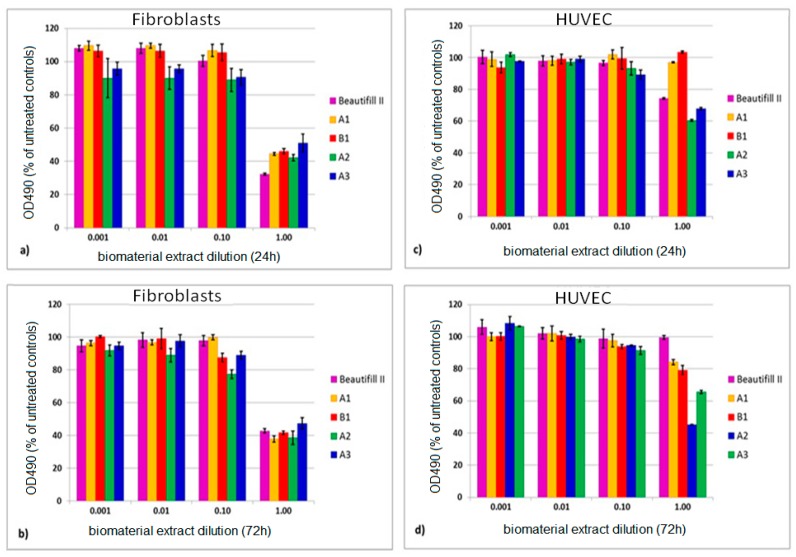
Comparative viability of fibroblasts exposed to the biomaterial extracts—24 h (**a**) and—72 h (**b**); human umbilical endothelial vein cultures (HUVEC) exposed to material extracts—24 h (**c**); and—72 h (**d**), respectively. Results are presented as % of the untreated controls absorbance at 490 nm; n = 3, average and SD are shown for each experiment.

**Table 1 materials-12-04021-t001:** The composition of the experimental giomer pastes.

No.	Resin				Hibrid Filler			
	Bis-GMA %	UDMA %	TEGDMA %	HEMA %	PRG1 %	PRG2 %	FHAP %	Radiopaque Glass %
A1	14	–	6	–	16	–	8	56
B1	14	–	6	–	-	16	8	56
A2	14	–	–	6	16	–	8	56
A3	–	14	–	6	16	–	8	56

**Table 2 materials-12-04021-t002:** The residual monomer related to the initial amount of the monomer in the sample.

Sample Code	Bis-GMA %	UDMA %	HEMA %	TEGDMA %
A1	1.62 ± 0.037	–	–	0.61 ± 0.166
B1	1.65 ± 0.050	–	–	0.69 ± 0.135
A2	2.13 ± 0.103	–	3.67 ± 0.169	–
A3	–	0.83 ± 0.071	1.17 ± 0.056	–
Beautifil II	1.08 ± 0.087 *	–	–	0.58 ± 0.137 *

Notes: For the calculation of the unreacted monomer in the case of Beautifil II sample, the composition from “Material Safety Data Sheet” [43] was taken into account (Bis-GMA 7.5% and TEGDMA 5% in the dental product).

**Table 3 materials-12-04021-t003:** The residual monomer related to the weight of the sample.

Sample Code	Bis-GMA %	UDMA %	HEMA %	TEGDMA %	Total Residual Monomer %
A1	1.13 ± 0.138	–	–	0.182 ± 0.019	1.312 ± 0.081
B1	1.18 ± 0.142	–	–	0.204 ± 0.013	1.384 ± 0.075
A2	1.49 ± 0.146	–	1.101 ± 0.057	–	2.591 ± 0.182
A3	–	0.829 ± 0.122	0.352 ± 0.073	–	1.181 ± 0.331
Beautifil II	0.65 ± 0.109	–	–	0.114 ± 0.232	0.764 ± 0.092

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
