# Peer review of "New Pre-reacted Glass Containing Dental Composites (giomers) with Improved Fluoride Release and Biocompatibility"

_materials, 2019, doi:10.3390/ma12234021_

Round 1

Reviewer 1 Report

The manuscript fits into the scope of the journal. The results are interesting and well presented, nevertheless there are some points to clarify. The major concern of the study is that the authors missed evaluating the functional and mechanical integrity of their experimental materials. No information about this important issue and it should be provided.   Page 4, line 167: the discs were polymerized before the evaluation of reacted monomers. After the polymerization how were stored the disc samples and for how much time? this could affect the amount of unreacted monomers.   page 4, line 171, why the authors extracted the monomers for 10h and not more? Please clarify and show the maximum concentration reached related to the maximum time extraction (plateau).    page 5, line 17: why the diameter of the sample discs were 15 mm and not 10 mm as in the samples used for the evaluation of unreacted monomers?   Page 6, line 234: morphological analysis must be integrated by an evaluation of the functional and mechanical proprieties of the new materials compared to the commercial one.

Author Response

Dear Reviewer,

Thank you for your evaluation of the manuscript and for your observations.

Below you have our answer to each of your observations.

Point 1:  The major concern of the study is that the authors missed evaluating the functional and mechanical integrity of their experimental materials. No information about this important issue and it should be provided.  

Response 1: Undoubtedly, the morphological, functional and mechanical integrity of a restorative material is an important issue. However, we set as the main purpose of this work to prepare a series of experimental giomers with increased fluoride release and to verify if these compositions have the required biocompatibility or even an improved biocompatibility.

Regarding the evaluation of the integrity of the experimental giomers developed by our group, we included in the “Introduction” section a paragraph presenting one of our previous study in which the marginal adaptation (that is related to the functional integrity) of an experimental giomer having a composition similar to the composition of A1 giomer from the present work in combination with two experimental two-step self-etch resin-modified glass-ionomer adhesive systems is evaluated and compared to the results obtained for Beautifil II and FL-Bond II adhesive.

Line 100: “The marginal adaptation was evaluated by determining the microleakage of an experimental giomer in combination with two experimental two-step self-etch resin-modified glass-ionomer adhesive systems and compared the results with the ones obtained for Beautifil II and FL-Bond II adhesive The first original primer contained as main component the polyalkenoic acid based on acrylic acid (AA), itaconic acid (IA) modified with polymerizable groups, while the second one contained a modified polyalkenoic acid which contained a rest of N-acryloyl-L-leucine besides the AA and IA. The original giomer contains a pre-reacted glass based also on the second mentioned AA/IA/N-acryloyl-L-leucine copolymer. The study was conducted using forty-two box-type class V standardized cavities which were prepared on the facial and oral surfaces of each tooth. After performing the clinical protocol, the restored teeth underwent 500 thermocycles in an LTC 100 thermocycling device (LAM Technologies Electronic Equipment, Firenze, Italy) that had two water baths at 5°C and 550C. The dye penetration score and the percentage of the dye penetration length were determined. A lower microleakage value was recorded for the adhesive system containing the AA/IA/N-acryloyl-L-leucine copolymer grafted with methacrylic groups compared with the commercial adhesive.” [Hodisan, I.; Prejmerean, C.; Buruiana, T.; Prodan, D.; Colceriu, L.; Petean, I.; Furtos, G.; Prejmerean, V.; Cotisel, M.T. Innovative Adhesive Systems for Dental Giomer Restorations. Rev. Chim.-Bucharest. 2018, 69, 2693-2702].

Point 2: Page 4 line 167: the discs were polymerized before the evaluation of reacted monomers. After the polymerization how were stored the disc samples and for how much time? this could affect the amount of unreacted monomers.  

Response 2: We completed the sentence in the manuscript :

Line: 183 “The disc samples (10 mm in diameter and 1mm in height) were light cured, weighed and immediately subjected to the extraction of the residual monomers in 50 mL chloroform at 600C for 10 hours.”

Point 3: page 4, line 171, why the authors extracted the monomers for 10h and not more? Please clarify and show the maximum concentration reached related to the maximum time extraction (plateau).   

Response 3: We propose this method in order to evaluate as precisely as possible the amount of unreacted monomer (mainly from the chemical point of view), of course the amount of extracted monomers in water, or artificial saliva will be much smaller.

From our experiments, the plateau is reached after 8 hours of extraction using these specific conditions. However, we subject the materials to extraction for a longer time (10 hours) every time.

We used this method in our previous articles, such as: “Cristina Prejmerean, Tinca Buruiana, Teresa Nunes, Marioara Moldovan and Loredana Colceriu (2011). Biocomposites Based on New Monomer Systems Reinforced with Micro / Nanoparticles and Glass Fibers, Metal, Ceramic and Polymeric Composites for Various Uses, John Cuppoletti (Ed.), ISBN: 978-953-307-353-8, InTech, Available from: http://www.intechopen.com/articles/show/title/biocomposites-based-on-new-monomer-systems-reinforced-with-micro-nanoparticles-and-glass-fibers]”. We include this reference in the present work as reference [35]

Line 185: This method of extraction of the unreacted monomer from dental composites was developed by our group and used in certain previous studies [35 ].” 

The maximum concentration reached for the monomers are presented in Tables 2 and 3, related to the maximum time extraction (10 hours). We completed the sentence in the manuscript:

Line 292: “The percents of the Bis-GMA, UDMA, HEMA and TEGDMA unreacted monomers extracted in chloroform at 600C for 10 hours (the maximum concentration being reached after 8 hours under these conditions) related to the initial amount of the corresponding monomer in the sample and to the weight of the sample, respectively are shown in Table 2 and Table 3, respectively”

Point 4: page 5, line 17: why the diameter of the sample discs were 15 mm and not 10 mm as in the samples used for the evaluation of unreacted monomers?  

Response 4: Yes, you are right. We could use the same sample size in both cases.  However both residual monomer determination and fluoride release determination are not standardized methods, different authors using different sizes for the specimens. As we presented in the manuscript, the dimensions of samples/ storage media used by different authors in the determination of the fluoride release varied very much . Line 464 “….the different sizes of samples (between 4 mm and 10 mm in diameter), the different amounts of storage media (between 1 ml and 15 ml deionized water)…. “. We consider that in both cases, namely determination of the residual monomer and determination of fluoride release, the ratio between the diameter of the sample (10 or 15 mm) and the volume of the solvent (50 ml chloroform or 45 ml deionized water) was enough for the extraction of the elutable species to occur under optimal conditions.

Point 5: Page 6, line 234: morphological analysis must be integrated by an evaluation of the functional and mechanical proprieties of the new materials compared to the commercial one.

Response 5: Following your recommendations, we include the determination of the flexural strengths (in accordance with ISO 4049/2000) of the investigated giomers as distinct paragraphs in the paper in the “Materials and methods”, “Results” and “Discussion” sections. We replaced the SEM images of the fracture appearance of the broken giomers from the “Results” section with the fracture images of the giomers after the flexural strength determinations.

All the modifications appear in blue in the revised manuscript.

Reviewer 2 Report

Thank you for the manuscript. The paper is interesting and well written.

Here are some minor issues the authors should address:

Line 80-81: Please, add reference to the statement.

Please, specify the units used in Table 1

Line 155: please, specify the wavelength and light intensity used as well as give the curing time

Please, explain the reason behind performing SEM analysis of giomers’ surface and fracture morphology. The aim of the study involved the biocompatibility and fluoride release from experimental giomers. No mention of the SEM analysis results was made in the discussion.

Line 161: SiC papers polished, rather than sanded the samples

Please, explain why the cell cultures: HDFa and HUVEC were used.

Fig 5: please, add detailed descriptions for y-axis in the charts

Please, give explanation to all abbreviations used in the manuscript

Author Response

Dear Reviewer,

Thank you for your evaluation of the manuscript and for your recommendations

Here are some minor issues the authors should address.

Below you have our answer to each of your observations.

Point 1: Line 80-81: Please, add reference to the statement.

Response 1: We added the references [1] and [2]. 

Point 2:  Please, specify the units used in Table 1

Response 2: We specified the units (%) in Table 1.

Point 3: Line 155: please, specify the wavelength and light intensity used as well as give the curing time

Response 3: We modified in the text:

Line 174: “For the obtaining of hardened giomer samples, the giomer pastes were exposed for 20 seconds to a visible radiation having the wavelength in the range of 470 nm and the intensity of 950 mW cm-2 generated by LED.E dental lamp (Guilin Woodpecker Medical Instruments Co.).“

Point 4: Please, explain the reason behind performing SEM analysis of giomers’ surface and fracture morphology. The aim of the study involved the biocompatibility and fluoride release from experimental giomers. No mention of the SEM analysis results was made in the discussion.

Response 4: At your suggestion we introduced a distinct paragraph “4.4. SEM investigations of giomers” in the “Discussion” section of the paper, where we explained the reason behind performing the SEM investigations:

Line 546: ‘We investigated the surface and fracture morphology of the experimental giomers A1, B1, A2 and A3 to compare it with the one of the commercial product Beautifil II in order to detect any differences in size, shape, distribution or amount of the filler particles that could influence the properties of the investigated giomers, in particular the fluoride release and flexural strength”…..

Line 572:…”the smaller reinforcement of the experimental giomers, which allows a greater segmental mobility of the polymer chain can contribute to the greater release of fluoride ions in the case of the experimental giomers [ Fujimoto ]”.

We replaced the SEM images of the broken giomer samples with the SEM images of the fracture appearance after the flexural strengths determination, following the observations of another reviewer. We also, included the determination of flexural strengths (in accordance with ISO 4049/2000) of the investigated giomers as distinct paragraphs in the paper at the “Materials and methods”, “Results” and “Discussion” sections.

Point 5: Q: Line 161: SiC papers polished, rather than sanded the samples

Response 5 :We replaced in the text : ”All samples were SiC papers polished using……….”

Point 6: Please, explain why the cell cultures: HDFa and HUVEC were used.

Response 6: Line 257: “The cytotoxicity assay evaluated the overall effect of the biomaterials on normal, human cell lines in vitro. Human dermal fibroblasts, as well as mouse 3T3 fibroblasts [37] or L929 fibroblasts [38] were previously employed for dental materials testing [39, 40, 41], since the cytotoxic effect is similar to that exerted on oral fibroblasts, while the HUVECs [42] were used for comparison, because they are fetal cells, that can mimic the stem cells behavior when exposed to eluted substances from the tested biomaterials.

Point 7: Fig 5: please, add detailed descriptions for y-axis in the charts

Response 7: The name of each cell line and incubation time required to obtain the biomaterial extract was added to each chart in figure 6. The cell viability was calculated as % of the untreated controls, namely cell cultures exposed only to medium.

Point 8: Please, give explanation to all abbreviations used in the manuscript

Response 8: We explained the abbreviations which we omitted to explain in the manuscript:

poly(methyl methacrylate) (PMMA)

High Performance Liquid Chromatography (HPLC).

polytetrafluoroethylene (PTFE) filters

Point 9: English language and style are fine/minor spell check required 

Response 9: We checked the English language

All the modifications appear in blue in the revised manuscript.

Reviewer 3 Report

This is an interesting and well written paper needing some minor improvements:

page 2 line 73: “fluoride ion” – delete “ion” because it is redundant: fluoride already denotes the F- ion.

This recent paper on dental biomaterials could be added in the Introduction for the Reader’s benefit:

Biomaterials, current strategies, and novel nano-technological approaches for periodontal regeneration. Journal of Functional Biomaterials 2019;10:3

SEM images: please remove the black box on the bottom and insert a clear and near scale bar along with the magnification only.

Author Response

Dear Reviewer,

Thank you for your evaluation of the manuscript and for your recommendations.

This is an interesting and well written paper needing some minor improvements.

Below you have our answer to each of your observations:

Point 1 : page 2 line 73: “fluoride ion” – delete “ion” because it is redundant: fluoride already denotes the F- ion.

Response 1: We deleted “ion”. 

Point 2: This recent paper on dental biomaterials could be added in the Introduction for the Reader’s benefit:

Biomaterials, current strategies, and novel nano-technological approaches for periodontal regeneration. Journal of Functional Biomaterials 2019;10:3

Response 2: We included the reference in the paper, as reference [18]:

Line 693: 18. Iviglia, G.; Kargozar, S.; Baino, F. Biomaterials, current strategies, and novel nano-   technological approaches for periodontal regeneration. J. Funct. Biomater. 2019, 10, 1-36.

Point 3: SEM images: please remove the black box on the bottom and insert a clear and near scale bar along with the magnification only.

Response 3: We removed the black box on the bottom of the SEM images and inserted a clear and near scale bar along with the magnification only.

All the modifications appear in blue in the revised manuscript.

Reviewer 4 Report

The authors did not mention how many times they have repeated the in vitro experiments, nor they mention in the results if an average of values has been calculated. Double check of English language is required (e.g. cytotoxicity and not citotoxicity). 

Author Response

Dear Reviewer,

Thank you for your evaluation of the manuscript and for your recommendations.

Below you have our answer to each of your observations:

Point 1: The authors did not mention how many times they have repeated the in vitro experiments, nor they mention in the results if an average of values has been calculated. Double check of English language is required (e.g. cytotoxicity and not citotoxicity).

Response 1: Line 274: Each experiment was done in triplicate. Results are presented as the average of the percents of untreated control absorbance (OD490). Untreated cultures exposed to medium were used as controls.

At your observation we completed in the "Results" section with the maximum and minimum average values obtained for the cell viability.

Line 370: “When the undiluted extracts were used, the viability of the fibroblasts ranged between 32.27% (Beautifil II) and 52.02% (A3) after 24 hours of extraction, and between 37.88% (A1) and 47.70 (A3) after 72 hours of extraction. When HUVEC were used, the viability ranged from 60.40% (A2) to 99.34% (B1) after 24 hours and from 45.22% (A2) to 99.57% (Beautifil II) after 72 hours of extraction.”

We checked the English language.

All the modifications appear in blue in the revised manuscript.

Round 2

Reviewer 1 Report

The authors addressed all the concerns raised by the referee. The manuscript is now acceptable for the publication